# Tissue Inhibitor of Matrix Metalloproteinases-1 (TIMP-1) and Pulmonary Involvement in COVID-19 Pneumonia

**DOI:** 10.3390/biom13071040

**Published:** 2023-06-26

**Authors:** Maria Antonella Zingaropoli, Tiziana Latronico, Patrizia Pasculli, Giorgio Maria Masci, Roberta Merz, Federica Ciccone, Federica Dominelli, Cosmo Del Borgo, Miriam Lichtner, Franco Iafrate, Gioacchino Galardo, Francesco Pugliese, Valeria Panebianco, Paolo Ricci, Carlo Catalano, Maria Rosa Ciardi, Grazia Maria Liuzzi, Claudio Maria Mastroianni

**Affiliations:** 1Department of Public Health and Infectious Diseases, Sapienza University of Rome, Piazzale Aldo Moro 5, 00185 Rome, Italyclaudio.mastroianni@uniroma1.it (C.M.M.); 2Department of Biosciences, Biotechnologies and Environment, University of Bari “Aldo Moro”, 70121 Bari, Italygraziamaria.liuzzi@uniba.it (G.M.L.); 3Department of Radiological, Oncological and Pathological Sciences, Policlinico Umberto I, Sapienza University of Rome, Viale Regina Elena 324, 00161 Rome, Italyfrancoiafrate@gmail.com (F.I.); valeria.panebianco@uniroma1.it (V.P.); paolo.ricci@uniroma1.it (P.R.); carlo.catalano@uniroma1.it (C.C.); 4Infectious Diseases Unit, Santa Maria Goretti Hospital, Sapienza, University of Rome, 04100 Latina, Italy; 5Department of Neurosciences Mental Health and Sensory Organs, Sapienza University of Rome, 00161 Rome, Italy; 6Medical Emergency Unit, Sapienza University of Rome, Policlinico Umberto I, 00161 Rome, Italy; g.galardo@policlinicoumberto1.it; 7Department of Specialist Surgery and Organ Transplantation “Paride Stefanini”, Policlinico Umberto I, Sapienza University of Rome, 00161 Rome, Italy; f.pugliese@uniroma1.it; 8Unit of Emergency Radiology, Policlinico Umberto I, Sapienza University of Rome, Viale del Policlinico 155, 00161 Rome, Italy

**Keywords:** matrix metalloproteinases, MMP-2, MMP-9, sCD163, sCD14, zymography, ELISA, computed tomography, pulmonary fibrosis, post-acute sequelae of SARS-CoV-2 infection, PASC

## Abstract

**Background:** The aim of the study was to longitudinally evaluate the association between MMP-2, MMP-9, TIMP-1 and chest radiological findings in COVID-19 patients. **Methods:** COVID-19 patients were evaluated based on their hospital admission (baseline) and three months after hospital discharge (T post) and were stratified into ARDS and non-ARDS groups. As a control group, healthy donors (HD) were enrolled. **Results:** At the baseline, compared to HD (n = 53), COVID-19 patients (n = 129) showed higher plasma levels of MMP-9 (*p* < 0.0001) and TIMP-1 (*p* < 0.0001) and the higher plasma activity of MMP-2 (*p* < 0.0001) and MMP-9 (*p* < 0.0001). In the ARDS group, higher plasma levels of MMP-9 (*p* = 0.0339) and TIMP-1 (*p* = 0.0044) and the plasma activity of MMP-2 (*p* = 0.0258) and MMP-9 (*p* = 0.0021) compared to non-ARDS was observed. A positive correlation between the plasma levels of TIMP-1 and chest computed tomography (CT) score (ρ = 0.2302, *p* = 0.0160) was observed. At the T post, a reduction in plasma levels of TIMP-1 (*p* < 0.0001), whereas an increase in the plasma levels of MMP-9 was observed (*p* = 0.0088). **Conclusions:** The positive correlation between TIMP-1 with chest CT scores highlights its potential use as a marker of fibrotic burden. At T post, the increase in plasma levels of MMP-9 and the reduction in plasma levels of TIMP-1 suggested that inflammation and fibrosis resolution were still ongoing.

## 1. Introduction

COVID-19 pneumonia is characterized by diffuse alveolar damage and the infiltration of monocytes, macrophages, and lymphocytes into the pulmonary interstitium, blocking alveolar gas exchange and inducing acute respiratory distress syndrome (ARDS) [1]. Recently, neutrophilia has been shown to be a poor outcome predictor in COVID-19 [2,3]. Thus, the tissue infiltration by neutrophils into pulmonary capillaries and their extravasation into the alveolar space has been demonstrated in the autopsy of patients [4,5]. There is also growing interest in neutrophil extracellular traps (NET) in containing the virus and whether this could contribute to the development of thrombosis and ARDS, as typically seen in critically ill COVID-19 patients [6,7]. To date, there has been no specific treatment for severe acute lung injury (ALI) and ARDS due to COVID-19, and its management has been mostly supportive. It is crucial to better understand the pathophysiological processes activated by inflammatory mediators such as cytokines and matrix metalloproteinases (MMPs) with the aim of understanding their subsequent inhibition during complex treatment.

MMPs, a group of zinc-dependent endopeptidases, have the ability to degrade proteins in the extracellular matrix (ECM) [8]. MMPs are produced by many cell types, including lymphocytes and granulocytes, but are especially activated macrophages [9]. They support the migration of immune cells into infection sites and are further involved in a variety of endogenous pro-inflammatory and vasoactive cytokine responses, as well as coagulation and fibrinolysis cascades [10,11]. MMP activity is essential in several physiological processes, such as growth and wound healing, but also in inflammatory and vascular pathophysiology (e.g., tissue remodeling, arteriosclerosis) [10]. Moreover, their activity is regulated in vivo by specific endogenous tissue inhibitors of metalloproteinases (TIMPs) [12,13,14], and the balance between MMPs and TIMPs is important in maintaining the integrity of ECM [13]. In most underlying conditions that lead to a fatal outcome, critically ill COVID-19 patients seem prone to a systemic inflammatory cytokine storm [15], and, among the enzymes secreted, MMPs and their natural tissue inhibitors TIMPs have been induced and involved in this systemic inflammatory cascade [16,17].

In the lungs, the normal expression of MMPs and their TIMPs is tightly regulated, showing upregulation in their initial development, remodelling in response to tissue injury, and host defence against pathogens [18]. Every type of MMP, and the fluctuation in their levels, play a specific role in different lung diseases [19]. Indeed, as reported by Busceti et al. [20], in pulmonary embolism, high plasma levels of MMP-2, MMP-9, and neutrophil gelatinase-associated lipocalin (NGAL), a protein involved in the modulation of MMP-9 activity, were found.

The severe form of COVID-19 has many similar features to sepsis [21], and both MMP-2 and MMP-9 have been considered potential biomarkers for septic patients [22,23]. The gene expression of MMP-9 is upregulated in COVID-19 patients [24], and plasma levels of MMP-9 are directly proportional to the risk of respiratory failure [25]. A prognostic that is relevant to MMP-9 plasmatic levels was proposed by Duda et al. [22]. Additionally, MMP-9 plasmatic levels have been shown to be increased in severe COVID-19 and to be associated with mortality in those patients [26]. Moreover, despite the anti-inflammatory role of MMP-2 in septic patients, it has already been shown that MMP-2 is decreased [23]. Similarly, D’Avila-Mesquita et al. [27] showed the downregulation of MMP-2 in severe COVID-19 patients and an independent correlation with mortality, suggesting its potential role as a prognostic predictor in COVID-19. However, da Silva-Neto et al. [28] demonstrated, for the first time, that the MMP-2 activity and level increased significantly in the lung microenvironment of COVID-19 intubated patients.

Finally, Brusa et al. [29] showed the involvement of TIMP-1 in the fibrotic process, identifying TIMP-1 as a useful marker of the fibrotic burden and disease prognosis in patients with COVID-19 at an initial diagnosis.

Despite the current advances in understanding COVID-19 disease, there is still a considerable knowledge gap in the management of long-term sequelae in such patients, especially concerning pulmonary fibrosis. This study sought to investigate the association between MMP-2, MMP-9, and TIMP-1 and chest radiological findings. This approach could contribute to a better understanding of COVID-19 pathogenesis as well as the post-acute sequelae of SARS-CoV-2 infection (PASC), providing potential new therapeutic options for the future.

## 2. Materials and Methods

### 2.1. Study Population

COVID-19 pneumonia patients admitted to the Department of Public Health and Infectious Diseases, Policlinico Umberto I Hospital, Sapienza, University of Rome, were enrolled. As previously described [30,31,32], COVID-19-related pneumonia was diagnosed by a chest computed tomography (CT) scan that is associated with SARS-CoV-2 RNA detection from a nasopharyngeal swab through a commercial reverse transcription-polymerase chain reaction (RT-PCR) kit, following the manufacturer’s instructions (RealStar^®^ SARS-CoV-2 Altona Diagnostic, Hamburg, Germany).

On hospital admission, clinical information, and routine laboratory exams, including demographics, respiratory parameters with arterial oxygen partial pressure/fraction of inspired oxygen (PaO_2_/FiO_2_, P/F) ratio, lactate dehydrogenase (LDH), C-reactive protein (CRP), ferritin, D-dimer, and blood lymphocyte counts were collected. During routine laboratory tests, blood samples were collected (baseline). According to the clinical outcome, COVID-19 patients were stratified into two groups: ARDS and non-ARDS. ARDS was defined according to the 2012 Berlin criteria [24].

In addition, after three months of discharge, a subgroup of COVID-19 patients was evaluated in the post-COVID-19 surgery, and blood samples were collected and analysed (T post).

Finally, as a control group, healthy donors (HD) without known lung disease, matched for age and sex, with a negative nasopharyngeal swab for SARS-CoV-2 RNA detection and undetectable anti-SARS-CoV-2 specific IgG without any symptoms, were enrolled.

### 2.2. Chest CT and Image Analysis

As previously described [30], a semi-quantitative CT severity score was provided for all patients, considering the extent of involvement per each of the 5 lobes, as follows: 0, no involvement; 1, <5% involvement; 2, 5–25% involvement; 3, 26–50% involvement; 4, 51–75% involvement; and 5, >75% involvement. The resulting global CT score was the sum of each individual lobar score (0 to 25).

### 2.3. Microfluidic Next Generation Enzyme-Linked Immunosorbent Assay (ELISA)

During routine clinical testing, peripheral whole blood was collected in heparin-coated BD Vacutainer Blood Collection tubes (BD Biosciences, Franklin Lakes, NJ, USA). The blood samples were centrifuged within three hours of collection. The top layer (plasma) was harvested and stored at −80 °C for retrospective analysis. In the collected samples, the evaluation of MMP-9, TIMP-1, sCD163, and sCD14 plasma levels was assessed using the Simple Plex^TM^ Ella Assay (ProteinSimple, San Jose, CA, USA) on the Ella^TM^ microfluidic system (Bio-Techne, Minneapolis, MN, USA) according to the manufacturer’s instructions. As previously described [33,34], Ella^TM^ was calibrated using the in-cartridge factory standard curve. The limits of detection for MMP-9, TIMP-1, sCD163, and sCD14 were 10.5 pg/mL, 0.34 pg/mL, 318 pg/mL, and 1.0 pg/mL, respectively. The limits of detection were calculated by adding three standard deviations to the mean background signal determined from multiple runs.

### 2.4. Gelatinase Activity by Zymography

As previously described [35,36], the evaluation of plasma MMP-2 and MMP-9 activity was performed using SDS-PAGE zymography with gels copolymerized with gelatine, which represented a suitable substrate for the separation and visualisation of both MMP-2 and MMP-9 in the same run from a single biological sample. The main advantage of gelatine gel zymography is that, depending on the samples analysed and the experimental conditions used, it allowed the visualisation of both the latent and active forms of gelatinases, which could be identified by their molecular weight.

Briefly, 10 µL of each plasma sample was solubilized in 100 µL of a loading buffer containing SDS, then 10 µL of diluted plasma was applied on 10% polyacrylamide gels (10 cm × 10 cm), which had been copolymerized with 0.1% (*w*/*v*) gelatine. Stacking gels contained 5.4% polyacrylamide. Electrophoresis was carried out at 4 °C for approximately 2 h at 100 V. After electrophoresis, the gels were washed for 2 × 30 min in 2.5% (*w*/*v*) Triton X-100 in 100 mM Tris-HCl, pH 7.4 (washing buffer) to remove SDS and reactivate the enzyme, which was then incubated for 24 h at room temperature in 100 mM Tris-HCl, pH 7.4 (developing buffer).

For the development of enzyme activity, the substrate in the gels was stained with Coomassie brilliant blue R-250 and de-stained in methanol/acetic acid/H_2_O. MMP-2 and MMP-9 were detected as white bands of digestion on the blue background of the gel and were identified by co-localisation on the zymogram with human MMP-2 or MMP-9 standards (ALEXIS Biochemicals, San Diego, CA, USA). Quantitation of MMP-2 and MMP-9 activity was performed using computerized image analysis (Image Master 1D, Pharmacia Biotech, Buckinghamshire, UK) through one-dimensional scanning densitometry (Ultroscan XL, Pharmacia Biotech). MMP activities were expressed as the optical density (OD) × mm^2^, representing the scanning area under the curves, which considered both the brightness and width of the substrate lysis zone.

### 2.5. Statistical Analysis

Analysis was performed with Prism 9 (GraphPad). A probability value <0.05 was considered statistically significant, and MMP-2, MMP-9, TIMP-1, sCD163, and sCD14 were examined as continuous variables. The patient characteristics were compared using Student’s *t*-test or chi-square for continuous and categorical variables, respectively. Continuous variables were expressed as the median and interquartile range (IQR) with the assumption of a normal distribution. Categorical variables were expressed as counts and percentages. The groups were then compared by Student’s *t*-test or the Mann–Whitney U-test, as appropriate. The nonparametric Kruskal–Wallis test with Dunn’s post-test was used for comparing the medians of ARDS and non-ARDS groups with HD. The nonparametric Wilcoxon test was used for the longitudinal evaluation comparing the baseline and T post. Finally, Spearman rank correlation analysis was used to assess the relation between plasma levels of MMP-9 and TIMP-1 with clinical data.

## 3. Results

### 3.1. Study Population

In this single-centre observational study, during the first wave of the COVID-19 pandemic, 129 COVID-19 patients (72 males, 57 females) with a median age (IQR) of 64 (55–77) years were enrolled. As the control group, 53 HD (30 males, 23 females) with a median age (IQR) of 60 (57–68) years were included in the study.

As reported in Table 1, 66.7% of enrolled COVID-19 patients had at least one coexisting illness, and the most prevalent of these were hypertension (40.3%), cardiovascular disease (21.7%), diabetes (17.9%) and pulmonary disease (17.9%). On hospital admission, the most common symptoms were fever (79.8%), cough (44.9%), and dyspnea (33.3%). Sputum production was uncommon (2.3%). The most prevalent treatment for COVID-19 diagnosis was hydroxychloroquine (69.8%).

During hospitalisation, 46.5% of COVID-19 patients developed ARDS (ARDS group), and this group was dominated, compared to the non-ARDS one, by a higher percentage of patients that died during hospitalisation (36.7% versus 2.9%, respectively; *p* < 0.0001) (Table 1).

On hospital admission, the ARDS group was characterized by a higher percentage of males compared to the non-ARDS one (73.3% versus 40.4%, respectively; *p* = 0.0002) as well as a higher percentage of COVID-19 patients with at least one comorbidity (78.3% versus 56.5%, respectively; *p* = 0.0095) (Table 1). Among these comorbidities, the percentage of patients with cancer was higher in the ARDS group compared to the non-ARDS one (18.3% versus 2.9%, respectively; *p* = 0.0063). Concerning symptoms on hospital admissions, the ARDS group was characterized by a higher percentage of patients who showed shortness of breath as a symptom (48.3% versus 20.3%, respectively; *p* = 0.0013) and a lower percentage of patients who showed myalgia and arthralgia compared to the non-ARDS one (13.3% versus 31.9%, respectively; *p* = 0.0206) (Table 1).

Regarding treatments in the ARDS group compared to the non-ARDS one, we observed a higher percentage of patients treated with enoxaparin (53.3% versus 31.9%, respectively; *p* = 0.0197), corticosteroids (45.0% versus 18.8%, respectively; *p* = 0.0021), and tocilizumab (60.0% versus 26.1%, respectively; *p* = 0.00002) (Table 1).

In the ARDS group, we observed a higher white blood cell (WBC) absolute count (*p* = 0.0414), neutrophils (N) absolute count (*p* = 0.0254), and neutrophil/lymphocyte ratio (NLR) (*p* = 0.0006), with a lower lymphocyte (L) absolute count (*p* = 0.0015) as well as higher inflammatory levels of markers such as CRP (*p* < 0.0001), LDH (*p* < 0.0001), ferritin (*p* = 0.0002) and D-dimer (*p* = 0.0006) compared to the non-ARDS group (Table 1). Finally, in the ARDS group, a lower P/F ratio compared to the non-ARDS one was observed (*p* < 0.0001), as well as a higher chest CT score (*p* < 0.00001) (Table 1).

### 3.2. Baseline Assessment of Plasma Levels of MMP-9 and TIMP-1 and Plasma Activity of MMP-2 and MMP-9

Using next-generation ELISA, the plasma levels of MMP-9 and TIMP-1 were evaluated in 129 COVID-19 patients and 53 HD. On hospital admission, COVID-19 patients showed significantly higher plasma levels of MMP-9 and TIMP-1 as well as a higher MMP-9/TIMP-1 ratio compared to HD (*p* < 0.0001, *p* < 0.0001 and *p* = 0.0299, respectively) (Figure 1A, Figure 1B and Figure 1C, respectively) (Table 2).

The ARDS group showed significantly higher plasma levels of MMP-9 and TIMP-1 compared to the non-ARDS group (*p* = 0.0339 and *p* = 0.0044, respectively) (Figure 1A,B, Table 2). Otherwise, between the ARDS and non-ARDS groups, no significant differences in the plasma levels of the MMP-9/TIMP-1 ratio were observed (Figure 1C, Table 2). Both the ARDS and non-ARDS groups showed significantly higher plasma levels of MMP-9 and TIMP-1 compared to HD (MMP-9: *p* < 0.0001 and *p* = 0.0002, respectively; TIMP-1: *p* < 0.0001 and *p* < 0.0001, respectively) (Figure 1A and Figure 1B, respectively) (Table 2). Finally, the ARDS group showed a higher plasma level of the MMP-9/TIMP-1 ratio compared to HD (*p* = 0.0189) (Figure 1C, Table 2).

Regarding zymography data, on the hospital admission of COVID-19 patients with a higher plasma activity, MMP-2 and MMP-9, compared to HD, were observed (*p* < 0.0001 and *p* < 0.0001, respectively) (Figure 1D and Figure 1E, respectively) (Table 2). The ARDS group showed significantly higher plasma activity of MMP-2 and MMP-9 compared to the non-ARDS group (*p* = 0.0258 and *p* = 0.0021, respectively) (Figure 1D and Figure 1E, respectively) (Table 2). Both the ARDS and non-ARDS groups showed significantly higher plasma activity of MMP-2 and MMP-9 compared to HD (MMP-2: *p* < 0.0001 and *p* < 0.0001, respectively; MMP-9: *p* < 0.0001 and *p* < 0.0001, respectively) (Figure 1D and Figure 1E, respectively) (Table 2). As shown in Appendix A, only two main lysis bands, corresponding to MMP-2 and MMP-9, were found in the gelatin zymographic gels, as evidenced by comigration with standard plasma.

Finally, on the hospital admission of COVID-19 patients, higher plasma levels of sCD163 and sCD14 compared to HD were found (*p* < 0.0001 and *p* < 0.0001, respectively) (Figure 1G and Figure 1H, respectively) (Table 2). The ARDS group showed significantly higher plasma levels of sCD163 and sCD14 compared to the non-ARDS group (*p* < 0.0001 and *p* < 0.0001, respectively) (Figure 1G and Figure 1H, respectively) (Table 2). Both the ARDS and non-ARDS groups showed significantly higher plasma levels of sCD613 and sCD14 compared to HD (sCD63: *p* < 0.0001 and *p* = 0.0020, respectively; sCD14: *p* < 0.0001 and *p* < 0.0001, respectively) (Figure 1G and Figure 1H, respectively) (Table 2).

Considering all COVID-19 patients in terms of hospital admissions, positive correlations between the plasma levels of MMP-9 and WBC absolute count (ρ = 0.4794, *p* < 0.0001) (Figure 2A), N absolute count (ρ = 0.5403, *p* < 0.0001) (Figure 2B), NLR (ρ = 0.4261, *p* < 0.0001) (Figure 2C) and CRP (ρ = 0.2335, *p* = 0.0132) (Figure 2D) were found. Otherwise, a negative correlation between the plasma levels of MMP-9 and the P/F ratio was observed (ρ = −0.1917, *p* = 0.0352) (Figure 2E).

Moreover, positive correlations between the plasma levels of TIMP-1 and the WBC absolute count (ρ = 0.2442, *p* = 0.0065) (Figure 2F), N absolute count (ρ = 0.2993, *p* = 0.0008) (Figure 2G), NLR (ρ = 0.3227, *p* = 0.0003) (Figure 2H), CRP (ρ = 0.2404, *p* = 0.0107) (Figure 2I), D-dimer (ρ = 0.3812, *p* < 0.0001) (Figure 2J), ferritin (ρ = 0.4312, *p* < 0.0001) (Figure 2K), LDH (ρ = 0.3031, *p* = 0.0008) (Figure 2L), and sCD163 (ρ = 0.2450, *p* = 0.0050) (Figure 2M) were found. Otherwise, a negative correlation between the plasma levels of TIMP-1 and the P/F ratio (ρ = −0.2484, *p* = 0.0060) was observed (Figure 2N). Finally, a positive correlation between the plasma levels of TIMP-1 positive and a chest CT score (ρ = −0.2302, *p* = 0.0160) (Figure 2O) was found.

### 3.3. Evaluation at Post-COVID Clinic

Seventy-eight COVID-19 patients were followed up 3 months after discharge (T post), and the differences, compared to the baseline, were evaluated. Among them, during the visit to the post-COVID clinic, sixty-four patients reported respiratory symptoms such as difficulty breathing or shortness of breath, a cough, and chest pain and underwent a chest CT follow-up.

At T post, a significant increase in the WBC and L absolute count compared to the baseline (*p* = 0.0009 and *p* < 0.0001, respectively) was observed (Figure 3A and Figure 3C, respectively) (Table 3). Otherwise, at the T post, a reduction in NLR compared to the baseline (*p* < 0.0001) was found (Figure 3D, Table 3). The plasma levels of CRP, D-dimer, ferritin, and LDH were decreased at the T post, compared to the baseline (*p* < 0.0001, *p* < 0.0001, *p* < 0.0001, and *p* < 0.0001, respectively) (Figure 3E, Figure 3F, Figure 3G and Figure 3H, respectively) (Table 3). Finally, at T post, a reduction in the CT score compared to the baseline was observed (*p* < 0.0001) (Figure 3I, Table 3).

The longitudinal evaluation of MMP-9 plasmatic levels showed an increase in the T post compared to the baseline (*p* = 0.0088) (Figure 4A, Table 3). Otherwise, at the T post, a significant reduction in plasma levels of TIMP-1 (*p* < 0.0001), sCD163 (*p* = 0.0202), and sCD14 (*p* = 0.0010) were observed (Figure 4B, Figure 4F and Figure 4G, respectively) (Table 3). In addition, a significant increase in the MMP-9/TIMP-1 ratio (*p* < 0.0001) was observed (Figure 4C, Table 3), as well as a significant increase in the plasma activity of MMP-2 (*p* < 0.0001) (Figure 4D, Table 3). Finally, no difference in the plasma activity of MMP-9 was found (Figure 4E, Table 3).

At both the baseline and T post, significantly higher plasma levels of MMP-9, TIMP-1, and the MMP-9/TIMP-1 ratio compared to HD were observed (MMP-9: *p* < 0.0001 and *p* < 0.0001, respectively; TIMP-1: *p* < 0.0001 and *p* = 0.0184, respectively; MMP-9/TIMP-1 ratio: *p* = 0.0774 and *p* < 0.0001, respectively) (Figure 4A, Figure 4B and Figure 4C, respectively) (Table 3). Similarly, at both the baseline and T post, the significantly higher plasma activity of MMP-2 and MMP-9, as well as plasma levels of sCD163 and sCD14 compared to HD were observed (MMP-2: *p* < 0.0001 and *p* < 0.0001, respectively; MMP-9: *p* < 0.0001 and *p* < 0.0001, respectively; sCD163: *p* < 0.0001 and *p* = 0.0044, respectively; sCD14: *p* < 0.0001 and *p* < 0.0001, respectively) (Figure 4D, Figure 2E, Figure 2F, and Figure 2G, respectively) (Table 3).

## 4. Discussion

The main finding of this study is that the plasma levels of TIMP-1 were positively correlated to the chest CT score. CT is widely considered to be the best imaging modality to assess parenchymal abnormalities in COVID-19, and many radiological studies have provided a CT-based scoring system to quantify the extent of lung damage on a 25-point scale [37]. To our knowledge, this is the first study showing that plasma levels of TIMP-1 correlate with lung abnormality by radiological investigations.

COVID-19 starts as a respiratory disease that can progress to pneumonia, SARS, and multi-organ failure. Growing evidence has suggested that COVID-19 is a systemic illness that primarily injures the vascular endothelium, yet its underlying mechanisms remain unknown. COVID-19-related pneumonia with severe respiratory failure is also characterized by enhanced fibrosis and ECM apposition [38]. Furthermore, lung fibrosis is one of the major long-term complications in patients with COVID-19 [39].

Macrophages play an important role in the immune response in acute and chronic inflammation, and they participate in the degradation and remodelling of ECM. These cells can modulate the matrix turnover by synthesizing and secreting several MMPs and counter-regulatory inhibitors. TIMPs inhibit the activity of fully competent MMPs and decrease MMP precursor activation [40].

Herein, we investigated the plasma levels of MMP-9, TIMP-1 as well as the plasma activity of MMP-2 and MMP-9 in a cohort of well-characterized patients with COVID-19 at an acute stage of the disease and after three months from hospital discharge. In order to assess whether MMPs and TIMPs could be used as biomarkers of COVID-19 progression and lung fibrosis, a comparison between COVID-19 patients and HD without known lung disease, as well as between patients with ARDS and non-ARDS, was performed. Additionally, the obtained finding was correlated to plasma levels of sCD163 and sCD14, which are known as markers of monocyte/macrophage cells and clinical data.

At first, we demonstrated that MMP-2, MMP-9, and TIMP-1 were elevated among COVID-19 patients and associated with COVID-19 severity, which is consistent with altered extracellular matrix remodelling. In fact, some authors previously reported that MMPs play a key role in lung disease [41,42]. However, in these studies, the principal attention was on MMPs, and elevated TIMP-1 was interpreted in the context of MMP inhibition.

TIMP-1 promote fibroblast and other cell proliferation, showing antiapoptotic and proinflammatory effects [43,44]. TIMP-1 is expressed in a multiplicity of cells and is implicated in acute inflammatory reactions [45,46] while its expression is stimulated by different substances, including platelet-derived growth factors, the basic fibroblast growth factor, epidermal growth factor, and is elevated in hyperoxia-induced lung injury [47,48]. Specifically, platelet-derived growth factors and basic fibroblast growth factors showed a profibrotic effect in acute lung injury [49,50]. Metzemaekers et al. [51] reported significantly higher plasma levels of TIMP-1 in COVID-19 patients at intensive care unit (ICU) admission.

MMP-2 was synthesized by a wide variety of cells, including fibroblasts, endothelial cells, and alveolar epithelial cells, and played a role in lung inflammatory diseases [52,53]. The lack of MMP-2 reduced cellular infiltration and fibrosis in allotransplant models, whereas a deficiency in MMP-2 increased the susceptibility of mice to lethal asphyxiation in an asthma model [54,55]. In COVID-19 patients, da Silva-Neto et al. reported that pro-MMP-2, detected by zymography, was extremely associated with non-survival COVID-19 patients [28].

MMP-9, which is present in low quantities in healthy lungs, is abundant in lung diseases characterized by tissue remodelling, such as asthma, pulmonary fibrosis, and chronic obstructive pulmonary disease (COPD) [40]. In ALI, the MMP-9 released from neutrophils promoted the inflammation and degradation of the alveolar–capillary barrier, further stimulating the migration of inflammatory cells and the destruction of lung tissue [39]. In lung tissue from COVID-19 patients, the MMP-9 gene was up-regulated, and protein contributed to cytokine recruitment [56].

In line with all these authors, we observed higher plasma levels of MMP-9 and TIMP-1 in COVID-19 patients who developed ARDS. Intriguingly, the plasma activity of MMP-2 and MMP-9 was higher in these patients. In contrast to the data reported by D’Avila-Mesquita et al. [27], in our cohort of COVID-19 patients, the higher plasma activity of both MMP-2 and MMP-9 on admission was found. It is plausible to speculate that the different result in plasma MMP-2 activity was due to the time at which the blood samples were taken. Indeed, the authors measured the plasma activity of MMPs within 48 h of ICU admission. Conversely, in our study, we collected blood samples on hospital admission. In addition, it was necessary to specify that only two main digestion bands, corresponding to MMP-2 and MMP-9, were detectable in our gelatin zymography gels. In fact, in the experimental conditions used, we were unable to discriminate between the pro-from and the active form of MMPs. On the other hand, even when both the active form and the pro-form of MMP-2 and MMP-9 were detectable on gelatin zymography gel, it was not possible to know how much of the band in the active form could have been endogenously active, i.e., not bound to his endogenous inhibitor TIMP [57]. Indeed, during zymography, run TIMPs were chemically dissociated from MMPs; therefore, the amount of active MMP that was not inhibited by binding to TIMP could not be accounted for in this analysis [57].

Finally, as already reported [58,59,60], we confirmed the increase in plasma levels of sCD163 and sCD14 on the hospital admission of COVID-19 patients, especially in those who developed ARDS, highlighting their potential use when assessing the risk of progression in the disease.

We found positive correlations between plasma levels of MMP-9 and WBC, N and NLR, and CRP. The involvement of MMP-9 in inflammatory diseases, such as its high plasma levels in COVID-19 patients from our study, was also supported by our previous demonstration that the plasma MMP-9 level was significantly higher in patients with community-acquired pneumonia compared with that in control subjects [61]. In line with other authors, we observed positive correlations between the plasma levels of TIMP-1 and WBC, N, NLR, CRP, D-dimer, ferritin, and LDH, highlighting the peculiar aspect of the involvement of TIMP-1 in the fibrotic process: TIMP-1 indicated decreased collagen degradation and was a strong predictor of early fibrosis [29,62]. The positive correlation between plasma TIMP-1 and sCD163 observed in this study suggested the possible role of TIMP-1 as a trigger of monocyte activation in inflammatory diseases, as recently suggested by Eckfeld et al. [63].

Moreover, we observed negative correlations between the plasma levels of MMP-9 and TIMP-1 with a P/F ratio. Similar to our findings, other studies have showed that respiratory failure, as defined by a P/F ratio, was negatively correlated with increased plasma MMP-9 and TIMP-1 [25,64]. This could be explained by increased oxidative stress in the groups of worse gas exchange. Indeed, TIMP-1 has been shown to be elevated in connection with hyperoxia in the lung [47,48,50] and in the course of oxidative damage to the venous endothelium [65]. On the other hand, TIMP-1 has functions that are independent of MMP inhibitory activity [43,45,46]. We speculated that the stimulation of the fibrotic process, the inhibition of proteolysis, and the promotion of proinflammatory processes by TIMP-1 could worsen the outcome of COVID-19. In addition, its antiapoptotic effect could inhibit the clearance of inflammatory processes. In addition, because of the antiapoptotic effect of TIMP-1, the inhibition of the clearance of inflammatory cells from the lung could be inhibited. This was also highlighted by the positive correlation between plasma TIMP-1 and the chest CT score, highlighting the potential role of TIMP-1 in the fibrotic process during COVID-19. Indeed, an exaggerated matrix accumulation and lack of matrix degradation could induce progressive lung remodelling or fibrosis, which could be related to the elevated expression of TIMP-1 in the lung [39].

Although we did not find significant differences in the plasma levels of MMP-9/TIMP-1 among the ARDS group compared to the non-ARDS one, we observed significantly higher levels only in the ARDS group compared to HD. A dynamic balance between the synthesis and degradation of ECM components was required for lung structure and functional maintenance. An increase in the MMP-9/TIMP-1 ratio was also associated with the severity of lung disease [66] and could indicate airway inflammation and bronchial injury while causing clinical differences in chronic airway diseases [67,68].

In the longitudinal investigation, we observed an increase in the plasma levels of MMP-9 and a reduction in the plasma levels of TIMP-1, sCD163, and sCD14. Thus, the plasma levels of the MMP-9/TIMP-1 ratio increased over time. Otherwise, the plasma activity of MMP-2 increased at the T post compared to the baseline, while the plasma activity of MMP-9 did not change. The persistent increase in plasma levels of MMP-9 and the plasma activity of MMP-2, as well as the steady plasma activity of MMP-9 found in this study, indicated the late role of these proteins in the inflammatory process, particularly during the repair phase, and could also be relevant to disease recovery, as already suggested by other authors [69]. Instead, the reduction in plasma levels of TIMP-1, sCD163, and sCD14 in line with the reduction in the chest CT score could be explained as an ongoing fibrosis resolution.

Overall, these results underscore the peculiar involvement of TIMP-1 in the fibrotic process by indicating decreased collagen degradation and assuming the role of a strong predictor of early fibrosis [62]. Our data indicate that TIMP-1 might be a useful marker of fibrotic damage and prognosis at the initial stage of COVID-19 pneumonia. Indeed, several molecular mechanisms involving the MMP pathway have been identified as relevant players in COVID-19 pneumonia [70]. Tissue damage during COVID-19 is associated with the activation of members of the MMP family [71,72]. The targeting MMP pathway has been proposed as a therapeutic strategy to counterbalance a host-marked pro-inflammatory response to COVID-19 [73]. In addition to being an MMP inhibitor, TIMP-1 is an independent pro-inflammatory and pro-growth factor [74,75,76]. Thus, the measurement of circulating TIMP-1 levels could be useful to assess the prognosis and adopt a personalized treatment approach.

Our study has some limitations. A larger sample size was needed to obtain a better assessment of TIMP-1 circulating levels as a prognostic biomarker in COVID-19 patients and to investigate its potential role in monitoring post-COVID symptoms. Moreover, we did not detect the plasma levels of MMP-2. Finally, the inability to discriminate between the pro-forms and the active forms of MMP-2 and MMP-9 in gelatin zymography gels did not allow us to analyse the ratio between proMMPs and active-MMPs.

In conclusion, our data showed that plasma levels of TIMP-1 in COVID-19 patients correlated with the severity of the disease as well as with inflammatory markers and the chest CT score, suggesting that TIMP-1 could serve as a non-invasive biomarker for prognosis and lung fibrosis in COVID-19. TIMP-1 and MMPs are worthy of further studies to assess their potential as prognostic and predictive biomarkers in COVID-19 patients. Finally, our longitudinal evaluation could contribute to clarifying the immune pathogenesis of COVID-19. Overall, these findings could help the clinical management of patients with COVID-19, improving the safety profile.

## Figures and Tables

**Figure 1 biomolecules-13-01040-f001:**
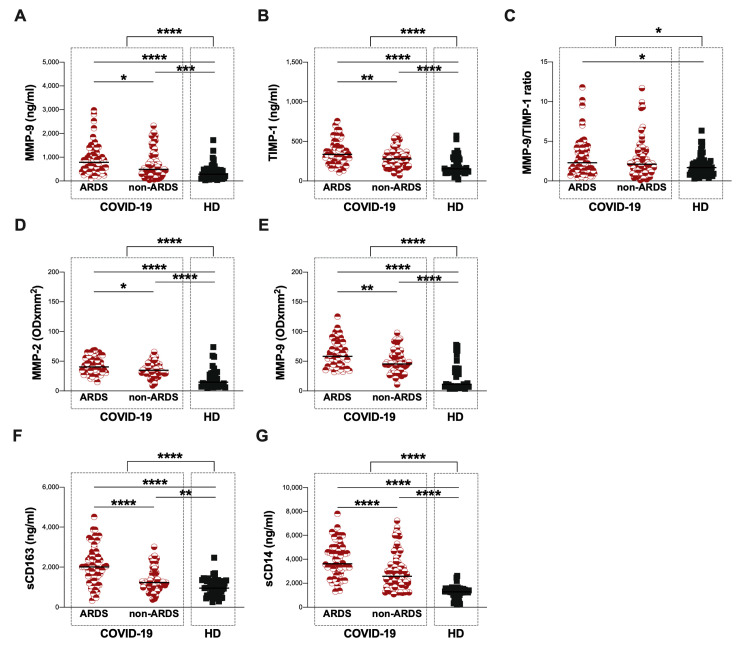
**Next-generation ELISA and gelatin zymography data at the baseline**. Plasma levels of MMP-9 (**A**), TIMP-1 (**B**), MMP-9/TIMP-1 ratio (**C**), sCD163 (**F**) and sCD14 (**G**), as well as the plasma activity of MMP-2 (**D**), and MMP-9 (**E**) were evaluated in COVID-19 patients (dashed line box on the left) and HD (dashed line box on the right), was stratified according to ARDS development. The differences between the total population of COVID-19 patients, and HD were evaluated using the nonparametric Mann–Whitney test. The differences between ARDS, non-ARDS groups and HD were assessed using the nonparametric Kruskal–Wallis test with Dunn’s post-test. Horizontal lines represent medians. ARDS: acute respiratory distress syndrome; HD: healthy donors; MMP-9: matrix metalloproteinase-9; TIMP-1: tissue inhibitor of metalloproteinase-1; MMP-2: matrix metalloproteinase-2; OD: optical density; sCD163: soluble CD163; sCD14: soluble CD14. *: 0.05 < *p* < 0.01; **: 0.01 < *p* < 0.001; ***: 0.001 < *p* < 0.0001; ****: *p* > 0.0001.

**Figure 2 biomolecules-13-01040-f002:**
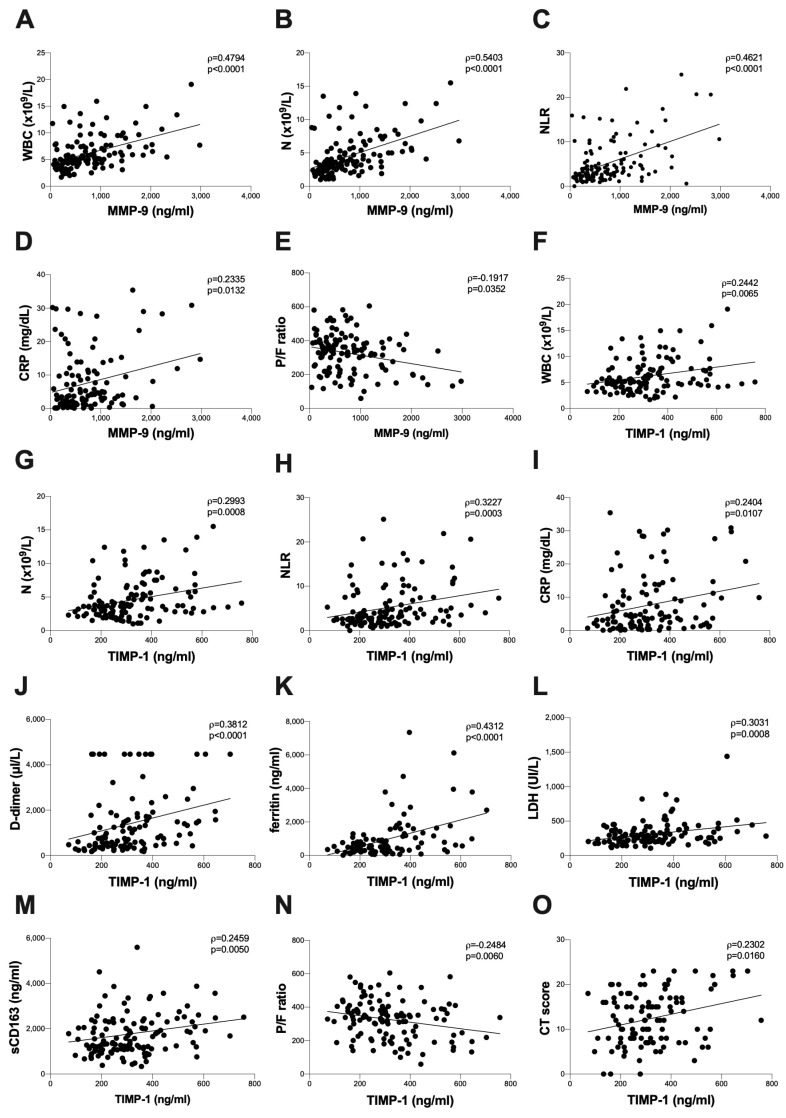
**Correlations between baseline plasma levels of MMP-9 and TIMP-1 with clinical data.** The correlation between levels of MMP-9 and TIMP-1 with the clinical parameters was evaluated in plasma samples from COVID-19 patients collected at the time of hospital admission using the regression test. A positive correlation was detected between plasma levels of MMP-9 and WBC (R^2^ = 0.2156, *p* < 0.0001) (**A**); neutrophil absolute count (R^2^ = 0.2449, *p* < 0.0001) (**B**); NLR (R^2^ = 0.4261, *p* < 0.0001) (**C**); and CRP (R^2^ = 0.6960, *p* = 0.0049) (**D**). A negative correlation was detected between plasma levels of MMP-9 and the P/F ratio (R^2^ = 0.06696, *p* = 0.0042) (**E**). A positive correlation was detected between plasma levels of TIMP-1 and WBC (R^2^ = 0.07493, *p* = 0.0022) (**F**); neutrophil absolute count (R^2^ = 0.08855, *p* = 0.0009) (**G**); NLR (R^2^ = 0.3227, *p* = 0.0003) (**H**); CRP (R^2^ = 0.2404, *p* = 0.0115) (**I**); D-dimer (R^2^ = 0.3812, *p* < 0.0001) (**J**); ferritin (R^2^ = 0.1833, *p* < 0.0001) (**K**); LDH (R^2^ = 0.09266, *p* = 0.0008) (**L**); sCD163 (R^2^ = 0.03857, *p* = 0.0257) (**M**); and chest CT score (R^2^ = 0.0.07575, *p* = 0.0038) (**O**). A negative correlation was detected between plasma levels of TIMP-1 and the P/F ratio (R^2^ = 0.05672, *p* = 0.0085) (**N**). All correlations were performed using the Spearman test. Spearman coefficient (ρ) and statistical significance (*p*) are reported in the graphics. MMP-9: matrix metalloproteinase-9; TIMP-1: tissue inhibitor of metalloproteinase-1; WBC: white blood cells; N: neutrophils; NLR: neutrophil/lymphocyte ratio; CRP: C-reactive protein; LDH: lactate dehydrogenase; sCD163: soluble CD163; P/F: arterial oxygen partial pressure/fraction of inspired oxygen; CT: computed tomography.

**Figure 3 biomolecules-13-01040-f003:**
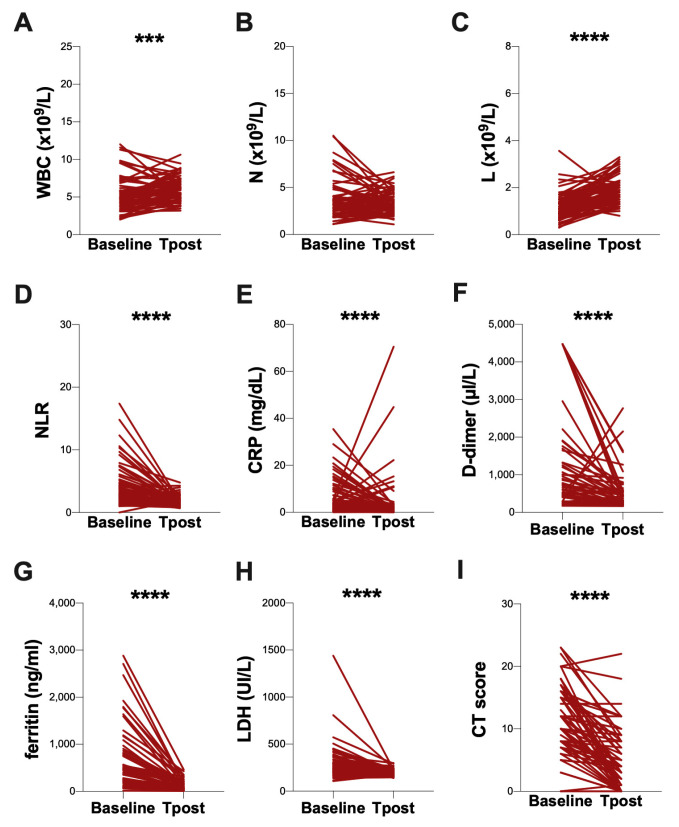
**Longitudinal evaluation of clinical data.** Longitudinal evaluation of WBC count (**A**), N absolute count (**B**), L absolute count (**C**), and NLR (**D**), as well as the plasmatic levels of CRP (**E**), D-dimer (**F**), ferritin (**G**), LDH (**H**) and the chest CT score (**I**), were performed using the Wilcoxon test. Baseline: on hospital admission; T post: three months from hospital discharge; WBC: white blood cells; N: neutrophils; L: lymphocytes; NLR: neutrophil/lymphocyte ratio; CRP: C-reactive protein; LDH: lactate dehydrogenase; CT: computed tomography. ***: 0.001 < *p* < 0.0001; ****: *p* > 0.0001.

**Figure 4 biomolecules-13-01040-f004:**
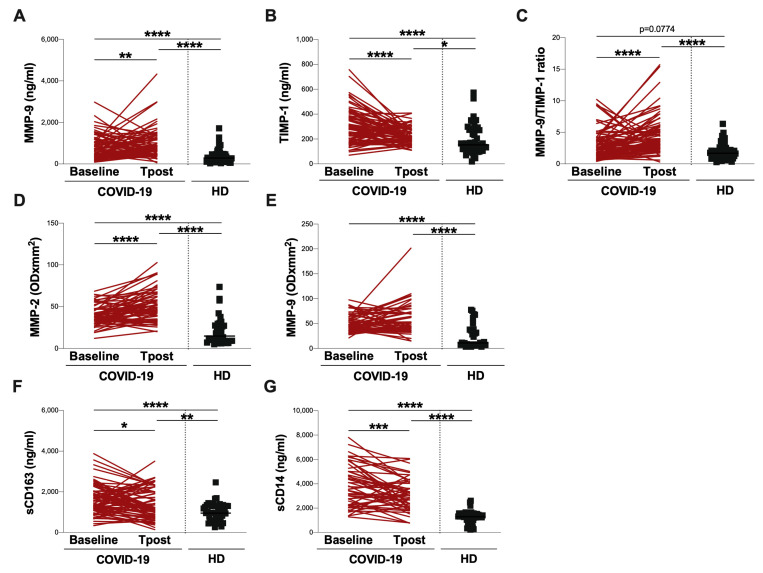
**Longitudinal evaluation of next-generation ELISA and gelatin zymography data.** Plasma levels of MMP-9 (**A**), TIMP-1 (**B**), MMP-9/TIMP-1 ratio (**C**), sCD163 (**F**), and sCD14 (**G**), as well as the plasma activity of MMP-2 (**D**) and MMP-9 (**E**), were evaluated in COVID-19 patients at two time-points: on hospital admission (baseline) and three months from hospital discharge (T post). The differences were evaluated using the nonparametric Wilcoxon test. Moreover, the differences between both time points and HD were assessed using the nonparametric Kruskal–Wallis test with Dunn’s post-test. Horizontal lines represent medians. ARDS: acute respiratory distress syndrome; HD: healthy donors; MMP-2: matrix metalloproteinase-2; MMP-9: matrix metalloproteinase-9; TIMP-1: tissue inhibitor of metalloproteinase-1; OD: optical density; sCD163: soluble CD163; sCD14: soluble CD14. * 0.05 < *p* < 0.01; ** 0.01 < *p* < 0.001; *** 0.001 < *p* < 0.0001; **** *p* > 0.0001.

**Table 1 biomolecules-13-01040-t001:** Demographic and clinical features of enrolled COVID-19 patients.

	COVID-19 (n = 129)	ARDS (n = 60)	non-ARDS (n = 69)	*p* Value *
Male/Female	57/72	44/16	28/41	0.0013
Age (years)	64 (55–77)	70 (63–81)	58 (50–66)	<0.0001
Deaths/Alive	24/105	22/38	2/67	<0.0001
Comorbidities				
Any	86	47	39	0.0407
Hypertension	52	27	25	ns
Cardiovascular	28	17	11	ns
Diabetes	23	13	10	ns
Pulmonary	23	12	11	ns
Cancer	13	11	2	0.0071
Renal	6	5	1	ns
Symptoms				
Fever	103	51	52	ns
Cough	58	28	30	ns
Shortness of breath	43	29	14	0.0026
Myalgia or arthralgia	30	8	22	0.0225
Diarrhea	16	6	10	ns
Anosmia and ageusia	7	2	5	ns
Sputum production	3	2	1	ns
COVID-19 treatment				
Lopinavir/ritonavir	32	16	16	ns
Hydroxychloroquine	90	42	48	ns
Azithromycin	70	35	35	ns
Enoxaparin	54	32	22	0.0339
Corticosteroids	40	27	13	0.0041
Tocilizumab	54	36	18	0.0004
Laboratory findings				
WBC (×10^9^/L)	5.3 (4.3–7.5)	5.8 (4.4–7.9)	4.9 (3.9–6.7)	0.0414
Neutrophils (×10^9^/L)	3.7 (2.5–5.8)	4.3 (3.1–6.9)	3.6 (2.3–5.0)	0.0254
Lymphocytes (×10^9^/L)	1.0 (0.7–1.4)	0.8 (0.6–1.3)	1.2 (0.8–1.6)	0.0015
NLR	3.4 (2.1–7.3)	4.4 (2.9–10.2)	2.7 (1.7–4.8)	0.0006
CRP (mg/dl)	4.1 (1.4–10.7)	9.8 (3.0–17.4)	2.8 (0.7–5.6)	<0.0001
D-dimer (µg/mL)	899 (443–1762)	1512 (630–3352)	590 (423–1319)	0.0004
Ferritin (ng/mL)	562 (278–1181)	999 (347–1800)	452 (234–712)	0.0002
LDH (U/L)	271 (206–373)	309 (255–436)	236 (198–303)	<0.0001
P/F ratio	343 (255–398)	293 (209–340)	384 (343–438)	<0.0001
Chest CT score (0–25)	12 (8–16)	15 (10–20)	10 (7–15)	<0.0001

n: number, ARDS: acute respiratory distress syndrome, WBC: white blood cells, NLR: neutrophil/lymphocyte ratio, CRP: C-reactive protein, LDH: lactate dehydrogenase, P/F: arterial oxygen partial pressure/fraction of inspired oxygen, CT: computed tomography, ns: not significant. Data are shown as median (IQR, interquartile range). * The 2-tailed X2 test or Fisher’s exact test and the nonparametric comparative Mann–Whitney test were used for comparing proportions and medians, respectively, between ARDS and non-ARDS COVID-19 patients.

**Table 2 biomolecules-13-01040-t002:** Next generation ELISA and zymography data in the study population at the baseline.

	COVID-19 (n = 129)	ARDS (n = 60)	non-ARDS (n = 69)	HD (n = 53)
Next generation ELISA				
MMP-9 (ng/mL)	606 (347–1067)	785 (401–1179)	489 (309–982)	287 (124–464)
TIMP-1 (ng/mL)	300 (206–386)	335 (345–437)	278 (191–361)	153 (115–273)
MMP-9/TIMP-1 ratio	2.2 (1.3–3.8)	2.3 (1.3–4.2)	2.1 (1.2–3.6)	1.7 (1.0–2.3)
sCD163 (ng/mL)	1553 (1110–2300)	2007 (1353–2559)	1226 (1005–1713)	952 (588–1300)
sCD14 (ng/mL)	3210 (2198–4548)	3717 (2920–4996)	2581 (2006–3557)	1294 (813–1511)
Zymography data				
MMP-2 (ODxmm^2^)	37 (30–48)	40 (32–53)	35 (29–45)	15 (9–28)
MMP-9 (ODxmm^2^)	49 (37–68)	58 (42–80)	46 (34–60)	11 (7–38)

Data are shown as the median and interquartile range (IQR). ARDS: acute respiratory distress syndrome; HD: healthy donors; MMP-9: matrix metalloproteinase-9; TIMP-1: tissue inhibitor of metalloproteinase-1; sCD163: soluble CD163; sCD14: soluble CD14; MMP-2: matrix metalloproteinase-2; OD: optical density.

**Table 3 biomolecules-13-01040-t003:** Longitudinal evaluation of clinical and next generation ELISA and zymography data.

	Baseline (n = 78)	T Post (n = 78)
Laboratory findings		
WBC (×10^9^/L)	4.9 (3.9–5.9)	5.6 (4.7–7.1)
Neutrophils (×10^9^/L)	3.3 (2.4–4.1)	3.2 (2.5–4.3)
Lymphocytes (×10^9^/L)	1.1 (0.8–1.4)	1.8 (1.5–2.3)
NLR	2.9 (1.9–4.8)	1.7 (1.4–2.2)
CRP (mg/dl)	3.2 (1.4–8.9)	1.1 (0.8–2.4)
D-dimer (µg/mL)	613 (411–1408)	325 (239–566)
Ferritin (ng/mL)	467 (227–885)	88 (41–188)
LDH (U/L)	252 (203–363)	188 (164–218)
Chest CT score (0–25)	12 (8–16)	4 (0–8)
Next generation ELISA		
MMP-9 (ng/mL)	558 (329–1031)	805 (465–1224)
TIMP-1 (ng/mL)	292 (201–382)	211 (177–272)
MMP-9/TIMP-1 ratio	2.3 (1.2–3.7)	3.6 (2.5–5.4)
sCD163 (ng/mL)	1542 (1121–2027)	1141 (797–2066)
sCD14 (ng/mL)	3564 (2235–5023)	3080 (2090–4084)
Zymography data		
MMP-2 (ODxmm^2^)	37 (31–47)	50 (35–65)
MMP-9 (ODxmm^2^)	48 (37–64)	50 (36–73)

Data are shown as the median (IQR, interquartile range). n: number; WBC: white blood cells; NLR: neutrophil/lymphocyte ratio; CRP: C-reactive protein; LDH: lactate dehydrogenase; CT: computed tomography; MMP-9: matrix metalloproteinase-9; TIMP-1: tissue inhibitor of metalloproteinase-1; sCD163: soluble CD163; sCD14: soluble CD14; MMP-2: matrix metalloproteinase-2; OD: optical density.

## Data Availability

All data generated or analyzed during this study are included in this published article. The datasets used and/or analyzed during the current study are available from the corresponding author on reasonable request.

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
