# Peer review of "Tissue Inhibitor of Matrix Metalloproteinases-1 (TIMP-1) and Pulmonary Involvement in COVID-19 Pneumonia"

_biomolecules, 2023, doi:10.3390/biom13071040_

Round 1

Reviewer 1 Report

The authors aimed to longitudinally evaluate the association between MMP-2, MMP-9 and TIMP-1 and chest radiological findings in COVID-19 patients.

- The study is very interesting and novel.

- The issue of MMPs, inflammation and COVID-19 is relevant.

- As MMP-9 activity is increased in inflammatory conditions, especially in pulmonary complications, by NGAL, I think this condition should be included in the introduction and/or the discussion. At this purpose cite and comment the article by Busceti et al. Pulmonary embolism, metalloproteinsases and neutrophil gelatinase associated lipocalin. Acta Phlebol. 2013. 14(3), 115–121.

- Methods and results section are well detailed.

- English should be revised for style.

English should be revised for style. It's a little hard to read.

Author Response

As suggested by Referee#1, in the “Introduction” section (page 2, lines 77-80), we commented and cited the article by Busceti et al., 2013.

Reviewer 2 Report

Maria Zingaropoli and co-authors aimed to describe the longitudinal evaluation  between MMP-2, MMP-9 and TIMP-1 and chest radiological findings in COVID-19 patients. Overall, there is a great cohort of patients and the study provide interesting details on proteases and a protease inhibitor. However, there are a few points that should be clarified:

1- In Figure 1 legend, the authors mentioned zymography and should be changed to gelatin zymography as there is also casein zymography.

2- How was the gelatin zymography data calculated?  The authors should show some examples on the gel as the levels of proMMP-9, MMP-9, proMMP-2 and MMP-2 can be detected but need to be visualized. Also, the authors report MMP-2 and MMP-9 as per their zymography data but no mention of proMMP-2 and proMMP-9 levels. What was the different in ratio between proMMPs and active MMPs? This is a key point that could impact the conclusions of the study.

3- Figure 1 and 4 figure legends, the spelling for MMP-2 and MMP-9 should be matrix metalloproteinase-2 and -9.

4- MMP-9 and TIMP-1 can be produced by many cells. What is the main source of MMP-9 and TIMP-1? What cells are producing it?

5-  What did they author did not look at the levels of TIMP-2? It is a key regulator of MMP2 and its activator, MT1-MMP (MMP14).

Minor comments:

Line 57: i would not use the word “very” as it might not sound scientific : it is  very important to better understand the pathophysiological processes”

Overall, the manuscript is well-written and the quality of the english is high.

Author Response

3- Figure 1 and 4 figure legends, the spelling for MMP-2 and MMP-9 should be matrix metalloproteinase-2 and -9.

As suggested by Referee#2, we correct the spelling for MMP-2 and MMP-9 in Figure 1 and Figure 4.

4- MMP-9 and TIMP-1 can be produced by many cells. What is the main source of MMP-9 and TIMP-1? What cells are producing it?

As suggested by the Referee#2, we reported in the “Introduction” section the main source of MMP-9 and TIMP-1 (page 2, lines 61-62).

5-What did they author did not look at the levels of TIMP-2? It is a key regulator of MMP2 and its activator, MT1-MMP (MMP14).

We thank the reviewer for his/her observation, but the multiplicity of tests performed to obtain the results shown in this manuscript did not allow us to also dose TIMP-2 levels, given the limited amount of sample available. The reason why we focused on TIMP-1 rather than TIMP-2 stems from the observation that among TIMPs, TIMP-1 is highly induced after triggered immune response whereas TIMP-2, TIMP-3 and TIMP-4 are constitutively secreted and modifications in their levels are lower than those of TIMP-1 (Leco et al. 1992; Manoury et al. 2006; Duda et al. 2020). In addition, altered TIMP-1 expression and MMP-9/TIMP-1 ratio was observed in COVID-19 patients with pulmonary involvement as well as in other viral pathologies involving fibrosis (Brusa et al. 2022; Guizani et al. 2021; Latronico et al. 2016). However, because a role of TIMP-2 has recently been hypothesized not only as an inhibitor of MMPs but also as a protein that can influence many signaling and inflammatory processes, investigation of its implication in the development of lung injury during COVID-19 will be a priority in our future studies.

Minor comments:

Line 57: i would not use the word “very” as it might not sound scientific: it is very important to better understand the pathophysiological processes”

As suggested by Referee#2, we rephrased the sentence (page 2, lines 57-59).

Comments on the Quality of English Language

Overall, the manuscript is well-written and the quality of the english is high

Maria Zingaropoli and co-authors aimed to describe the longitudinal evaluation between MMP-2, MMP-9 and TIMP-1 and chest radiological findings in COVID-19 patients. Overall, there is a great cohort of patients and the study provide interesting details on proteases and a protease inhibitor. However, there are a few points that should be clarified:

1- In Figure 1 legend, the authors mentioned zymography and should be changed to gelatin zymography as there is also casein zymography.

As suggested by the Referee#2, we changed the legend of Figure 1 (page 8, lines 261) and Figure 4 (page 12, line 334) as following “Figure 1. Next generation ELISA and gelatin zymography data at baseline” and “Figure 4. Longitudinal evaluation of next generation ELISA and gelatin zymography data”, respectively.

2- How was the gelatin zymography data calculated? The authors should show some examples on the gel as the levels of proMMP-9, MMP-9, proMMP-2 and MMP-2 can be detected but need to be visualized. Also, the authors report MMP-2 and MMP-9 as per their zymography data but no mention of proMMP-2 and proMMP-9 levels. What was the different in ratio between proMMPs and active MMPs? This is a key point that could impact the conclusions of the study.

As reported in the paragraph “Gelatinase activity by zymography” in Materials and Methods (page 3, lines 149-167), quantitation of MMP-2 and MMP-9 plasma activity was performed using computerized image analysis through one-dimensional scanning densitometry. Plasma MMP activities were expressed as optical density (OD) x mm2, representing the scanning area under the curve, which considers both brightness and width of the substrate lysis zone. MMP-2 and MMP-9 were identified by comparison with a standard plasma sample containing MMP-2 and MMP-9.

As evidenced in the zymographic gels reported below as example, in the analysis of our samples we only found MMP-2 and MMP-9. Therefore, the lack of the pro-forms of MMP-2 and MMP-9 does not allow us to analyze the ratio between proMMPs and MMPs.

Round 2

Reviewer 2 Report

The authors have made significant improvements on the manuscript.

However, there are still issues in reporting MMP2 and MMP9 activities. IN Figure 1 D and E and also Figure 4, how are the levels of proMMP2 and proMMP9 versus active MMP2 and MMP9? There are no examples of gelatin zymography gels. Gelatin zymography does not only report active MMP activity but by molecular weight can also present inactive proMMP2 and proMMP9 levels. Therefore, not 100% of these enzyme are likely active as measure by gelatin zymogrpahy. A ratio of pro vs active MMPs must be presented or reported to support the conclusions of MMP activity.

Author Response

As suggested by Referee#2, we included a supplementary figure (Supplementary Figure 1), showing two examples of our gelatin zymography gels, in which it is evident that in the samples analyzed and in the experimental conditions used, only two main digestion bands are present, corresponding to MMP-2 and MMP-9, as evidenced by the comigration with a standard plasma. A comment to Supplementary Figure 1 has been reported the “Results” section (page 7, line 256-258).  The inability to discriminate between the pro-forms and the active-form of MMP-2 and MMP-9 did not allow us to analyze the ratio between proMMPs and active-MMPs. On the other hand, as reported in the “Discussion” section (page 14, line 431-440), even when both the pro-form and the active form of MMP-2 and MMP-9 are discernible on the gelatin zymography gel it is not possible to know how much of the band of the active form would have been endogenously active, i.e. not bound to his endogenous inhibitor TIMP. Indeed, during the gelatin zymography run TIMPs are chemically dissociated from MMPs, therefore the amount of active MMP that is not inhibited by the binding to TIMP cannot be accounted for in this analysis. However, in the “Discussion” section (page 16, line 506-508), we specified that the inability to differentiate the pro-form from the active form of MMP-2 and MMP-9 represents a limitation of our study.

Round 3

Reviewer 2 Report

Thank for addressing my comments.